# The acute effect of metabolic cofactor supplementation: a potential therapeutic strategy against non-alcoholic fatty liver disease

Cheng Zhang[1,2,†] (ID), Elias Bjornson[3,4,†] (ID), Muhammad Arif[1,†], Abdellah Tebani[1] (ID), Alen Lovric[1,‡,§], Rui Benfeitas[1,¶] (ID), Mehmet Ozcan[1] (ID), Kajetan Juszczak[1], Woonghee Kim[1], Jung Tae Kim[1], Gholamreza Bidkhori[5], Marcus Ståhlman[3], Per-Olof Bergh[3] (ID), Martin Adiels[3], Hasan Turkez[6], Marja-Riitta Taskinen[7], Jim Bosley[8], Hanns-Ulrich Marschall[3], Jens Nielsen[4] (ID), Mathias Uhlén[1] (ID), Jan Borén[3,*] (ID) & Adil Mardinoglu[1,5,**] (ID)

## Abstract

The prevalence of non-alcoholic fatty liver disease (NAFLD) continues to increase dramatically, and there is no approved medication for its treatment. Recently, we predicted the underlying molecular mechanisms involved in the progression of NAFLD using network analysis and identified metabolic cofactors that might be beneficial as supplements to decrease human liver fat. Here, we first assessed the tolerability of the combined metabolic cofactors including L-serine, N-acetyl-L-cysteine (NAC), nicotinamide riboside (NR), and L-carnitine by performing a 7-day rat toxicology study. Second, we performed a human calibration study by supplementing combined metabolic cofactors and a control study to study the kinetics of these metabolites in the plasma of healthy subjects with and without supplementation. We measured clinical parameters and observed no immediate side effects. Next, we generated plasma metabolomics and inflammatory protein markers data to reveal the acute changes associated with the supplementation of the metabolic cofactors. We also integrated metabolomics data using personalized genome-scale metabolic modeling and observed that such supplementation significantly affects the global human lipid, amino acid, and antioxidant metabolism. Finally, we predicted blood concentrations of these compounds during daily long-term supplementation by generating an ordinary differential equation model and liver concentrations of serine by generating a pharmacokinetic model and finally adjusted the doses of individual metabolic cofactors for future human clinical trials.

**Keywords** L-serine, N-acetyl-L-cysteine (NAC), nicotinamide riboside (NR), and L-carnitine; NAFLD; systems medicine
**Subject Categories** Pharmacology & Drug Discovery; Computational Biology; Metabolism
**Mol Syst Biol. (2020) 16: e9495**

## Introduction

Hepatic steatosis (HS) is defined as the accumulation of large vacuoles of triglycerides in the liver ($> 5.5\%$ tissue weight) due to an imbalance between lipid deposition and lipid removal from the liver (Solinas et al, 2015; Francque et al, 2016; Samuel & Shulman, 2018). Non-alcoholic fatty liver disease (NAFLD) encompasses a broad spectrum of pathological conditions, ranging from simple HS to various degrees of liver inflammation such as non-alcoholic steatohepatitis

1  Science for Life Laboratory, KTH—Royal Institute of Technology, Stockholm, Sweden
2  School of Pharmaceutical Sciences, Zhengzhou University, Zhengzhou, China
3  Department of Molecular and Clinical Medicine, University of Gothenburg and Sahlgrenska University Hospital Gothenburg, Gothenburg, Sweden
4  Department of Biology and Biological Engineering, Chalmers University of Technology, Gothenburg, Sweden
5  Centre for Host-Microbiome Interactions, Faculty of Dentistry, Oral & Craniofacial Sciences, King's College London, London, UK
6  Department of Medical Biology, Faculty of Medicine, Atatürk University, Erzurum, Turkey
7  Research Programs Unit, Diabetes and Obesity, Department of Internal Medicine, Helsinki University Hospital, University of Helsinki, Helsinki, Finland
8  Clermont, Bosley LLC, Gothenburg, Sweden
   *Corresponding author. Tel: +46 31 342 2949; E-mail: jan.boren@wlab.gu.se
   **Corresponding author. Tel: +46 8 524 820 20; E-mail: adilm@scilifelab.se
   Lead Contact. Adil Mardinoglu
   †Joint first authors
   ‡Present address: Division of Clinical Physiology, Department of Laboratory Medicine, Karolinska Institutet, Karolinska University Hospital, Stockholm, Sweden
   §Present address: Unit of Clinical Physiology, Karolinska University Hospital, Stockholm, Sweden
   ¶Present address: Science for Life Laboratory, Department of Biochemistry and Biophysics, National Bioinformatics Infrastructure Sweden (NBIS), Stockholm University, Stockholm, Sweden

(NASH), which can progress to severe liver diseases, including cirrhosis and hepatocellular carcinoma (HCC). The prevalence of NAFLD continues to increase dramatically, and it has reached 25% at the population level (Estes *et al*, 2018) with the progressive epidemics of obesity and type 2 diabetes mellitus (T2DM). There are at present no approved effective medications for treating NASH.

In order to resolve the pathophysiology of NAFLD and to reveal the underlying molecular mechanisms involved in the progression towards NASH, a systems biology approach which enables integration and analysis of multi-layer omics data through the use of biological networks has been employed (Mardinoglu & Nielsen, 2015; Mardinoglu & Uhlen, 2016; Bosley *et al*, 2017; Mardinoglu *et al*, 2018a,b). Earlier, we generated a functional liver-specific genome-scale metabolic model (GEM) (Mardinoglu *et al*, 2014) and an integrated network (IN) (Lee *et al*, 2016) by merging GEMs with regulatory and protein–protein interaction networks. This integrative approach allows not only to reveal the key pathways, metabolites, and genes involved in the progression of liver diseases, but also to make solid predictions that can be experimentally tested due to the known regulatory effect of other proteins on metabolism.

Recently, we have combined clinical studies with stable isotopes, in-depth multi-omics profiling, and liver-specific networks to clarify the underlying mechanisms of NAFLD and develop strategies for prevention and treatment (Mardinoglu *et al*, 2017). Our integrative multi-tissue analysis has indicated that NAFLD patients have reduced *de novo* synthesis of glutathione (GSH) due to limited availability of serine and glycine, resulting in altered GSH and $NAD^+$ metabolism, which is a prevailing feature of NAFLD. To test the model-based predictions, we assessed the effect of short-term serine supplementation in NAFLD patients by providing an oral dose of ~ 20 g of L-serine (200 mg/kg) per day for 14 days showing that liver enzymes (ASAT, ALAT, ALP) and plasma triglycerides, as well as the amount of fat in liver were significantly decreased after supplementation of serine (Mardinoglu *et al*, 2017). Our model also indicated that supplementation of L-carnitine as well as precursors of GSH and $NAD^+$ including L-serine, *N*-acetyl-L-cysteine (NAC), and nicotinamide riboside (NR) would decrease liver fat accumulation by promoting the fat uptake and its oxidation in the mitochondria as well as generation of GSH required in the liver (Mardinoglu *et al*, 2017). These predictions were further tested in a mouse study and supplementation with such a formulation decreased the amount of liver fat (Mardinoglu *et al*, 2017).

Here, we first performed a 7-day rat toxicology to study tolerability of the combined metabolic cofactors and measured clinical parameters to identify potential side effects. Next, we performed a 5-day human calibration study by supplementing naturally occurring metabolic cofactors including 20 g L-serine, 3 g L-carnitine, 5 g NAC, and 1 g NR and a control study to reveal the acute global effect of the supplementation of combined metabolic cofactors by eliminating the effect of the fasting. We generated plasma metabolomics and inflammatory protein markers data to reveal the changes associated with the supplementation of these metabolic cofactors and measured the kinetics of the metabolites and proteins in the plasma of the healthy subjects. Next, we predicted altered pathways, reactions, and metabolites in liver due to the supplementation of the metabolic cofactors using metabolomics data and personalized genome-scale metabolic modeling. Moreover, we developed an ordinary differential equation (ODE) model to predict blood

concentrations of each metabolic cofactor during daily long-term supplementation and adjusted their doses. Finally, we analyzed literature data and the data generated in the supplementation study using pharmacokinetic (PK) modeling and statistical analysis.

# Results

### Seven-day oral (gavage) tolerability study in the rat

We performed a 7-day oral (gavage) toxicology study in the Wistar Hannover rats (without blinding) and assessed the tolerability of the combined metabolic cofactors including L-serine, NAC, NR, and L-carnitine tartrate (salt form of L-carnitine) at intended human clinical doses (Formulation I) and 10-fold (Formulation II) and 30-fold (Formulation III) dose levels. We assessed body weight, hematology, plasma chemistry, and gross pathology during the study.

Three groups of rats, each consisting of three females to acquire minimal required statistical power (without randomization), were given combined metabolic cofactors orally, once a day for 7 days. The dose levels administered are presented in Table EV1. From available data on the individual components, Formulation I was not expected to produce any adverse effects. In Formulations II and III, the levels of some of the components were approaching known tolerability levels; hence, adverse effects could be expected.

Several observations (e.g., plowing with the nose in the bedding material and excessive chewing) were made in all groups in connection with, or shortly after dosing, and are not considered of toxicological significance. Most severe observations, such as ataxia, cyanosis, irregular and/or respiration and decreased motor activity, were observed in Group 3 at Days 1 and 2. This led to a lowering of the dose (33%) in this group on Day 3. The new dose level was well tolerated for the remainder of the study. In all groups, some milder signs of discomfort (eyes half-shut and pilo-erection) were observed at Day 1, but were not present from Day 2 in Groups 1 and 2. This may only indicate a reaction to a new, unknown treatment, but may also indicate a tolerability buildup after repeated exposure.

We observed that none of the doses administered in this study caused any significant changes (Appendix Table S1) in hematological (Appendix Table S2) and plasma chemistry parameters (Appendix Table S3). We also did not detect any significant changes by the pathology analysis or during the macroscopic analysis at necropsy.

### Human calibration study with natural metabolic cofactors

We performed a 5-day calibration study by recruiting nine healthy male subjects without any medication (age 26–36 years, BMI 19.4–34.5 kg/m$^2$) to identify the acute global effect of metabolic cofactors supplementation (Fig 1A, Table EV2). The subjects stayed in the same hotel, had the same breakfast, and did not eat/drink anything until the end of the study in each day. We supplemented each of the four metabolic cofactors NR, L-carnitine, NAC, and L-serine to all subjects at separate days as well as the combined metabolic cofactors (i.e., a cocktail of the substances) at another day. Based on literature information, we supplemented 20 g L-serine, 3 g L-carnitine, 5 g NAC, and 1 g NR per day (Hurd *et al*, 1996; Hathcock & Shao, 2006; Garofalo *et al*, 2011). The study started at 8:00 every day and blood samples were collected before and 4 h after the

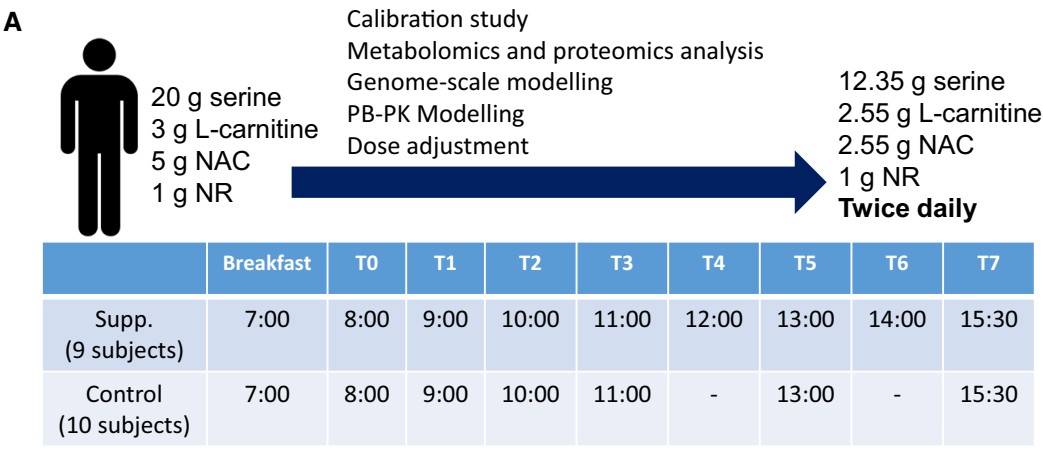

**A**

20 g serine
3 g L-carnitine
5 g NAC
1 g NR

Calibration study
Metabolomics and proteomics analysis
Genome-scale modelling
PB-PK Modelling
Dose adjustment

12.35 g serine
2.55 g L-carnitine
2.55 g NAC
1 g NR
**Twice daily**

| | Breakfast | T0 | T1 | T2 | T3 | T4 | T5 | T6 | T7 |
|---|---|---|---|---|---|---|---|---|---|
| Supp. (9 subjects) | 7:00 | 8:00 | 9:00 | 10:00 | 11:00 | 12:00 | 13:00 | 14:00 | 15:30 |
| Control (10 subjects) | 7:00 | 8:00 | 9:00 | 10:00 | 11:00 | - | 13:00 | - | 15:30 |

**B**

**Serine** — Metabolomics

**Carnitine** — Metabolomics

**Cysteine** — Metabolomics

**nicotinamide** — Metabolomics

**Figure 1.**

◀

**Figure 1.  Calibration study with the supplementation of metabolic cofactors.**

A   Summary of the metabolic cofactor supplementation and control study as well as the dosage of the metabolic cofactors before (left regimen) and after (right regimen) dosage adjustment based on this study.

B   Changes in plasma level of each cocktail substances in both supplementation and control studies (NR is detected in control study) compared to time baseline based on untargeted metabolomics measurement. The gray shaded area represents the 95% confidence level interval. For boxplots limits, the middle line represents the median. The upper and lower box limits represent the 25% quantiles. The upper and lower error bars correspond to 75% quantiles. The *P*-values are derived from one-way ANOVA (FDR < 0.05).

supplementation of individual metabolic cofactors. At the day we supplemented the combined metabolic cofactors, eight blood samples were collected during the day (Fig 1A). We measured the plasma level of glucose, insulin, free fatty acids, triglycerides, total cholesterol, HDL, LDL, and known liver markers including gamma GT, bilirubin, ASAT, ALAT, ALP before and after the study, and observed no significant differences (Table EV2). Subjects involved in the study did not report any side effect. Our analysis below focused on the day, where we supplemented the combined metabolic cofactors.

After 21 months, we asked the subjects involved in the study to run a follow-up control study and five of the subjects responded positively. We recruited another five healthy subjects and run a 1-day control study with 10 male subjects without any medication (age 26–36 years, BMI 19.8–35.8 kg/m$^2$) following the same study design in the metabolic cofactor supplementation study. Subjects stayed in the same hotel, had the same breakfast, and did not eat/drink anything until the end of the study. The study started at 8:00, and six blood samples were collected during the day after drinking a glass of water (Fig 1A).

**Plasma metabolites associated with the supplementation of metabolic cofactors**

We first quantified the plasma levels of ʟ-serine, ʟ-carnitine, and NAC using targeted metabolomics on the day we supplemented the combined metabolic cofactors and observed that supplementation of each metabolite increased the plasma levels proportionally (Appendix Fig S1 and Table EV3). We also generated untargeted metabolomics data using the plasma samples (Table EV4) and confirmed that plasma level of ʟ-serine, ʟ-carnitine, cysteine, and nicotinamide is increased after supplementation (Fig 1B). We observed a high degree of correlation between the serine (Pearson, *r* = 0.99) and ʟ-carnitine (Pearson, *r* = 0.95) levels detected using targeted and untargeted metabolomics analysis, as expected.

We have earlier shown that plasma levels of kynurenine, kynurenate, pyruvate, and ornithine are significantly positively associated with increased liver fat (Mardinoglu *et al*, 2017). We first investigated whether supplementation of the combined metabolic cofactors affected the plasma level of these substances (Table EV5). We found that plasma levels of kynurenine (Fig 2A), kynurenate (Fig 2B), pyruvate (Fig 2C), and ornithine (Fig 2D) were significantly decreased after the supplementation compared to baseline. In addition to the decrease in the plasma level of lipid structures (Appendix Figs S2 and S3), we also found that plasma levels of branch chain amino acids (BCAAs) including leucine (Fig 2E), isoleucine (Fig 2F), and valine (Fig 2G) which are significantly associated with insulin resistance and future incidence of T2D were significantly decreased after supplementation compared to baseline. In order to investigate if the decrease in the plasma levels of these

metabolites was associated with the supplementation of the metabolic cofactors, we performed a correlation analysis between these metabolites and supplemented metabolic cofactors (Table EV6, Appendix Figs S4 and S5). We found that the decreased plasma level of kynurenine is significantly negatively correlated with carnitine and NR; kynurenate is significantly negatively correlated with serine; pyruvate is significantly negatively correlated with NR and cysteine; and ornithine is significantly negatively correlated with serine (Fig 2H). We also observed that the decreased plasma level of BCAAs was significantly negatively correlated with plasma level of serine.

In order to eliminate the fasting effect on the kinetics of the plasma metabolites, we also generated untargeted metabolomics data using the plasma samples from the control study and detected the plasma level of 630 metabolites (Table EV7) of which 96 metabolites were detected in both studies (Table EV8). We observed that none of the significant correlations between the metabolites associated with increased liver fat and supplemented metabolic cofactors were identified in the control study.

Next, we studied the kinetics of the plasma metabolites by comparing to the baseline in the supplementation and control study (Fig 3A). We observed that the plasma levels of ʟ-serine, ʟ-carnitine, and cysteine were significantly different and showed different dynamics, as expected. For instance, we found that both the ʟ-serine and ʟ-carnitine levels kept almost constant in the control study which are opposite to their rapid increase exhibited in the supplementation study (Fig 1C). Interestingly, we found that the plasma cysteine level between the two studies has been shifted during the experiment, where it is significantly higher in the first 2 h but much lower in the last 2 h in the supplementation study (Fig 1C). This indicated that the supplementation not only boosted the cysteine level in the first half of the experiment, but also increased its consumption during the whole study. We also identified the metabolites showing the same and opposite trends in two studies. In addition to the plasma level of supplemented metabolic cofactors, directly associated metabolites to these cofactors including betaine, choline, glycine, cystine, carnitine derivatives; as well as other metabolites including citrulline, trimethylamine N-oxide (TMO), homoarginine, cortisone, and dimethylarginine showed opposite trend in two different studies.

Finally, to systematically evaluate the differences between two the studies and to eliminate the fasting effect, we compared the plasma levels of these metabolites between the subjects for each time point and identified 18 metabolites that were significantly (ANOVA test, adjusted *P* < 0.05) altered in at least three time points (Fig 3B). We observed the plasma level of citrulline (Fig 3C), TMO (Fig 3D), and dimethylarginine (Fig 3E) showed consistently significant differences. Citrulline is a key metabolite in the urea cycle, and its supplementation has been suggested to prevent the development of NAFLD (Sellmann *et al*, 2017). TMO is also significantly associated with NAFLD, and it is known as a potential plasma biomarker

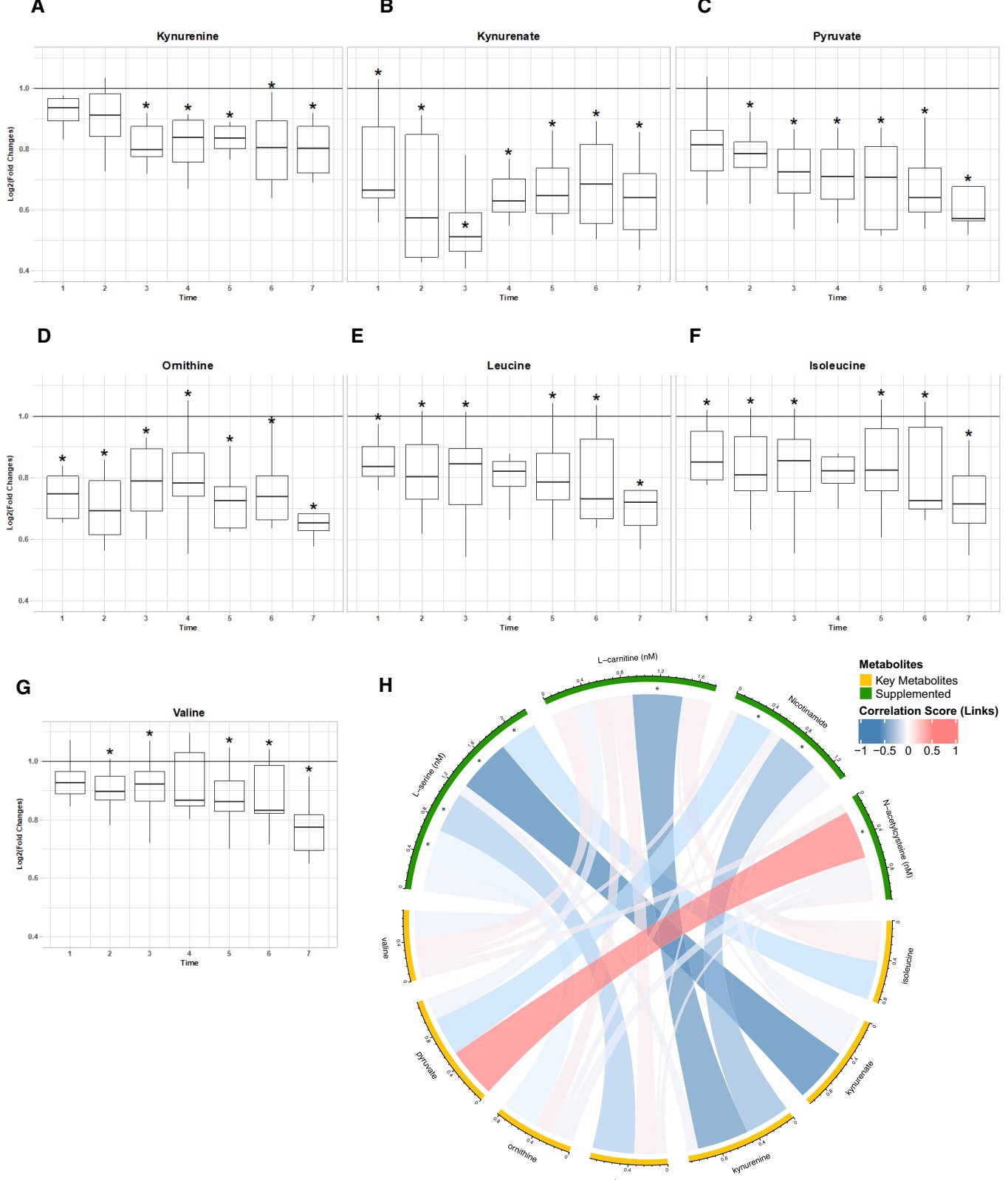

**Figure 2.　The changes in the plasma level of NAFLD associated metabolites due to the supplementation.**

A–G　Changes in plasma level of key metabolites compared to time baseline (solid line) based on untargeted metabolomics measurement in metabolic cofactor supplementation study. The upper and lower box limits represent the 25% quantiles. The upper and lower error bars correspond to 75% quantiles. *Denotes significance (one-way ANOVA; FDR < 0.05).

H　Spearman correlation (visualized by Circlize) between plasma levels of supplemented metabolic cofactors and key metabolites associated with high liver fat and insulin resistance. *Denotes significance (FDR < 0.05).

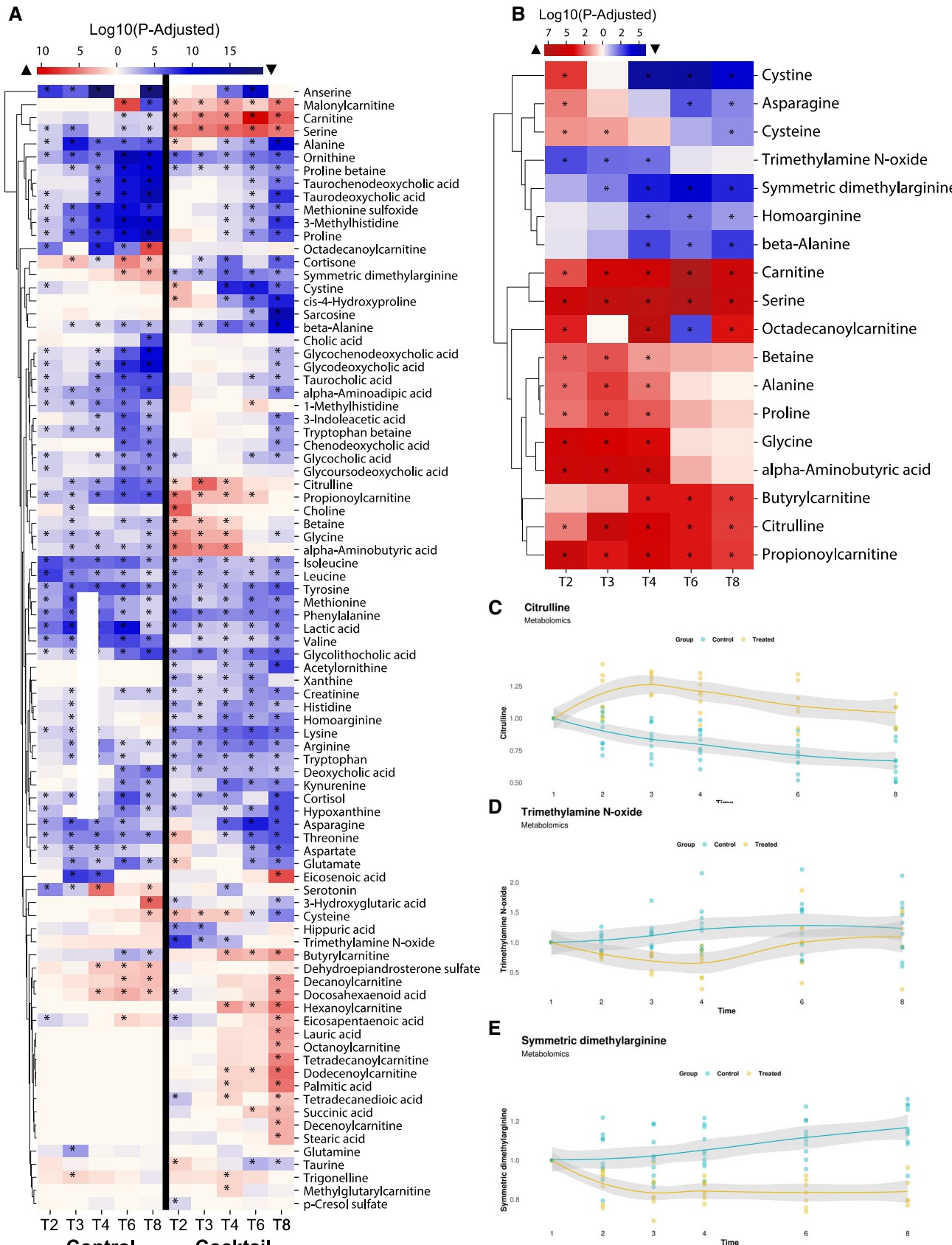

**Figure 3.**

---

**Figure 3. The effect of supplementation on plasma metabolites.**

A   A summary of significantly different plasma metabolites compared to time baseline in metabolic cofactor supplementation and control studies based on untargeted metabolomics measurements. Red and blue colors denote the increased and decreased plasma levels compared to baseline, respectively. *Denotes significance (one-way ANOVA; FDR < 0.05).

B   Metabolites significantly different between supplementation and control studies. Red and blue colors denote the increased and decreased plasma level in supplementation study compared to control study, respectively. *Denotes significance (one-way ANOVA; FDR < 0.05).

C–E   Changes in plasma level of key metabolites compared to time baseline based on untargeted metabolomics measurement in supplementation and control studies. The gray shaded area represents the 95% confidence level interval.

---

of NAFLD (Chen *et al*, 2016). The circulating dimethylarginine level is also related to NAFLD in previous studies (Kasumov *et al*, 2011). Taken together, global plasma metabolomics analysis based on supplementation and control study suggested that the supplementation of metabolic cofactors significantly improves the altered metabolism in NAFLD.

## Inflammatory protein markers associated with the supplementation of metabolic cofactors

We measured the plasma levels of 67 inflammation-related protein markers using proximity extension assay (PEA) proteomics-based inflammation panel (detected in more than 80% of the samples) in the supplementation (Table EV9) and control study (Table EV10). We observed significant reductions in plasma level of the inflammatory markers including fibroblast growth factor 21 (FGF21), FGF 19, osteoprotegerin (OPG), TNF receptor superfamily member 9 (TNFRSF9), C-C motif chemokine 23 (CCL23), CCL25, matrix metalloproteinase 1 (MMP1), MMP10, Fms-related tyrosine kinase 3 ligand (FLT3LG), TRANCE, interleukin 17C (IL-17C), IL-10, glial cell-derived neurotrophic factor (GDNF), and other inflammatory protein markers in the supplementation study compared to its baseline (Fig 4A).

We next measured the plasma levels of the protein markers in the control study. We found the plasma level of FGF19, MMP1, MMP10, FLT3L, TRANCE, CCL25, IL-10 is also significantly decreased compared to the baseline in the control study (Fig 4A), which suggested that their decrease in the supplementation study may be due to the fasting of the subjects.

To systematically evaluate the differences between two studies, we also compared the plasma levels of these inflammatory markers between the subjects for each time point and presented the markers that were significantly (ANOVA test, adjusted $P < 0.05$; Table EV11) altered in at least two time points (Fig 4B). We found that plasma level of four different markers including FGF21 (Fig 4C), OPG (Fig 4D), TNFRSF9 (Fig 4E), and CCL23 (Fig 4F) was significantly decreased with the supplementation after removing the effect of fasting.

Previous studies have shown that plasma level of FGF21 is elevated in NAFLD patients, and it is positively correlated with high liver fat in both mice and human (Dushay *et al*, 2010; Li *et al*, 2010,

2013; Rusli *et al*, 2016). The plasma level of FGF21 has also been suggested as a potential diagnostic marker of NAFLD. Consistent with these findings, we showed that supplementation of the metabolic cofactors significantly decreased the plasma level of FGF21 compared to baseline after eliminating the fasting effect. The rapid reduction in the plasma level of FGF21 agrees with our recent study where the plasma concentrations of FGF21 are decreased in parallel to the decrease in liver fat with a carbohydrate-restricted diet (Mardinoglu *et al*, 2018c).

There is also study reported that there is a link between TNFRSF9/CD137 and NAFLD in mice, showing that deficiency of TNFRSF9/CD137 prevents the mice from the development of NAFLD (Kim *et al*, 2011b). Hepatic cells are known to be able to produce OPG, and the plasma level of OPG has been shown to be positively correlated with liver fat content in dysmetabolic adults (Monseu *et al*, 2016). The plasma level of CCL23 has also been associated with plasma oxidative low-density lipoprotein (Kim *et al*, 2011a), which could affect the redox balance in hepatic cells, and the latter has been linked to hepatic steatosis in recent studies (Walenbergh *et al*, 2013). Finally, we observed that the plasma level of MMP1 showed significant differences due to the supplementation. The decreased plasma level of MMP1 has been associated with NAFLD in previous studies (Mahmoud *et al*, 2012; Schuppan *et al*, 2018). Hence, our analysis of plasma inflammation protein markers in the supplementation and control studies suggested that the supplementation may significantly affect the inflammation markers that are associated with NAFLD.

## Global effect in liver using liver GEMs

To investigate the potential global metabolic effect of the combined metabolic cofactors supplementation in liver, we integrated the untargeted metabolomics data and a comprehensive liver GEM, which is the representative of liver metabolism (Mardinoglu *et al*, 2014). Initial constraints in liver GEM were retrieved from measured liver flux data and literature reports (Felig *et al*, 1974; Hyotylainen *et al*, 2016; Mardinoglu *et al*, 2017). We set a mandatory ATP hydrolysis rate in the model to account for the resting energy expenditure of the organ that is calculated based on

---

**Figure 4. The effect of supplementation on plasma inflammatory markers.**

A   A summary of significantly different plasma inflammatory protein markers compared to time baseline in metabolic cofactor supplementation and control studies. Red and blue colors denote the increased and decreased plasma levels compared to baseline, respectively. *Denotes significance (one-way ANOVA; FDR < 0.05).

B   Proteins significantly different between supplementation and control studies. Red and blue colors denote the increased and decreased plasma level in supplementation study compared to control study, respectively. *Denotes significance (one-way ANOVA; FDR < 0.05).

C–F   Changes in the plasma level of key proteins compared to time baseline based on untargeted metabolomics measurement in supplementation and control studies. The gray shaded area represents the 95% confidence level interval. For boxplots limits, the middle line represents the median. The upper and lower box limits represent the 25% quantiles. The upper and lower error bars correspond to 75% quantiles. The *P*-values are derived from one-way ANOVA (FDR < 0.05).

---

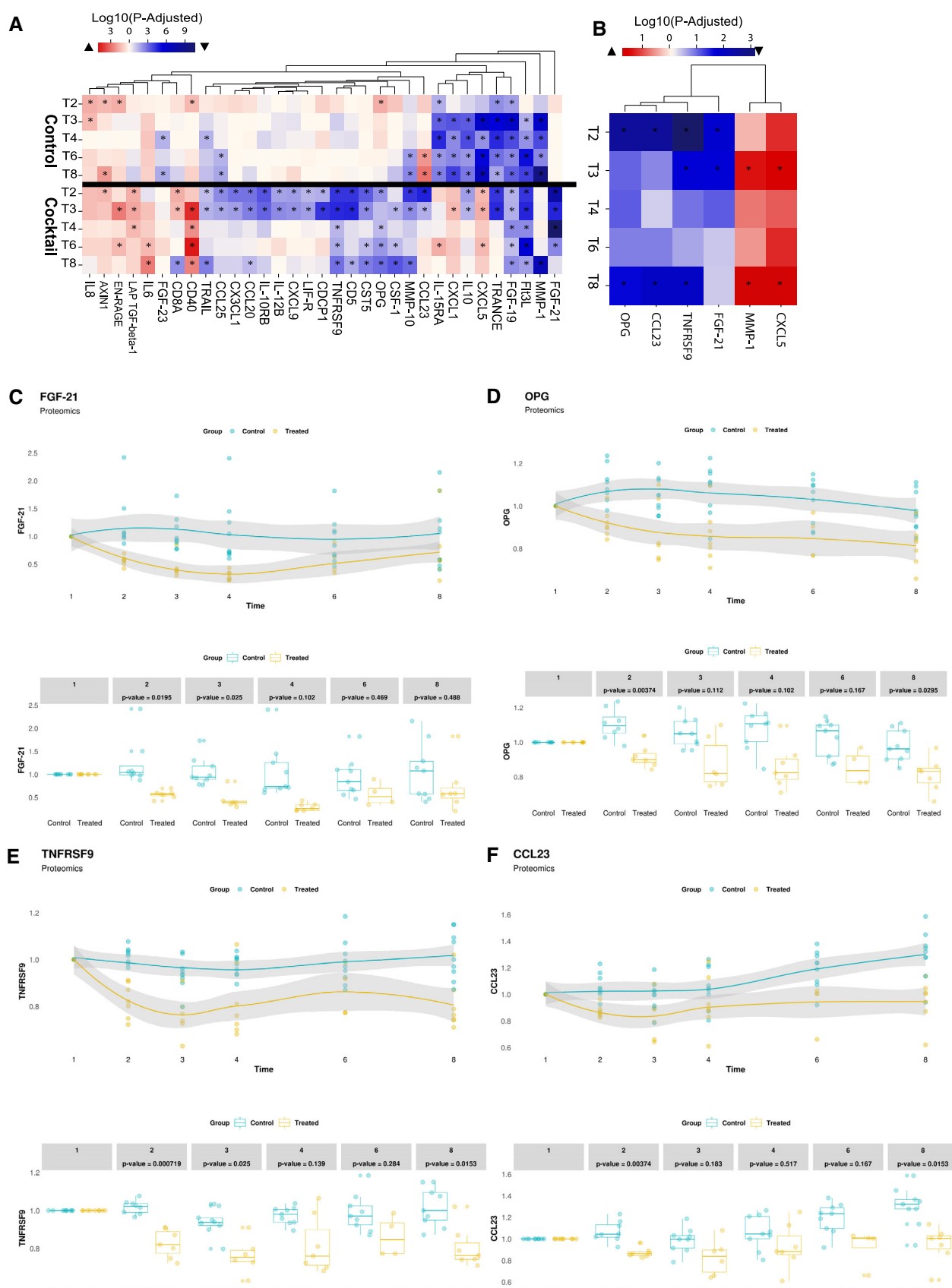

**Figure 4.**

previous experimental studies (McClave & Snider, 2001). We also incorporated the personalized changes in the plasma metabolite levels between the supplementation and control studies as additional constraints to the model for systematic interpretation of the metabolomics data. For instance, citrulline is increased in the plasma of the first subject with metabolic cofactor supplementation, and it is incorporated into the personalized model as either its consumption/secretion is decreased/increased in response to the supplementation. Moreover, we used oxidation of fat as an objective function since such supplementation can boost oxidation of fat in liver and eventually decrease the amount of fat in the liver. Finally, we predicted intracellular fluxes for five subjects who participated in both supplementation and control studies in a personalized manner using an adapted version of a previous method (Mardinoglu *et al*, 2015; Materials and Methods). The liver GEM used in this study together with all constraints and objective function is provided in Table EV12.

We performed the simulations for each individual involved in both studies, presented the model simulation results in Table EV13, and summarized the results in Fig 5A. First, we compared the fluxes of total fatty acid oxidation (set as an objective function) and found that these fluxes increased after combined metabolic cofactors supplementation compared to control study (Fig 5B). This prediction is in line with our assumptions and indicated that the supplementation of metabolic cofactors could promote oxidation of fat in liver. Second, we investigated the fluxes in glutathione synthesis and observed that flux carried by the reaction is increased due to the

supplementation of serine and NAC (Fig 5C–E). Moreover, we compared the global intracellular fluxes predicted by the model and observed increased uptake and oxidation of BCAAs (Fig 5F–K) and increased flux turnovers for both NAD$^+$ (Fig 5L) and L-carnitine (Fig 5M), indicating activation of mitochondria after the supplementation of metabolic cofactors. We also detected slight decrease in the fluxes carried by the reactions involved in glycolysis, indicating the inhibition of glucose metabolism (Fig 5N and O). Finally, we observed an increased flux of high-density lipoprotein secretion with supplementation of metabolic cofactors (Fig 5P and Q), which suggested another potential benefit effect of the supplementation in the liver fatty acids metabolism.

## Adjustment of the dose based on calibration study

Using the targeted metabolomics data obtained from the plasma of nine subjects, we developed a three-compartment ODE pharmacokinetic model to represent metabolic cofactors distribution within the body. The model structure was inferred from a mechanistic view of holdup in the stomach and absorption from the small intestine to the blood, with central clearance. The model was simultaneously fitted to the experimentally measured plasma concentrations for all subjects. Similar compartmental models for each metabolic cofactor were developed and fitted to measured plasma concentrations. Data used were from the nine subjects over the course of up to 24 h after supplementation. The models were fitted using a pooled-data approach, and bioavailability of each metabolic cofactor was set according to literature values.

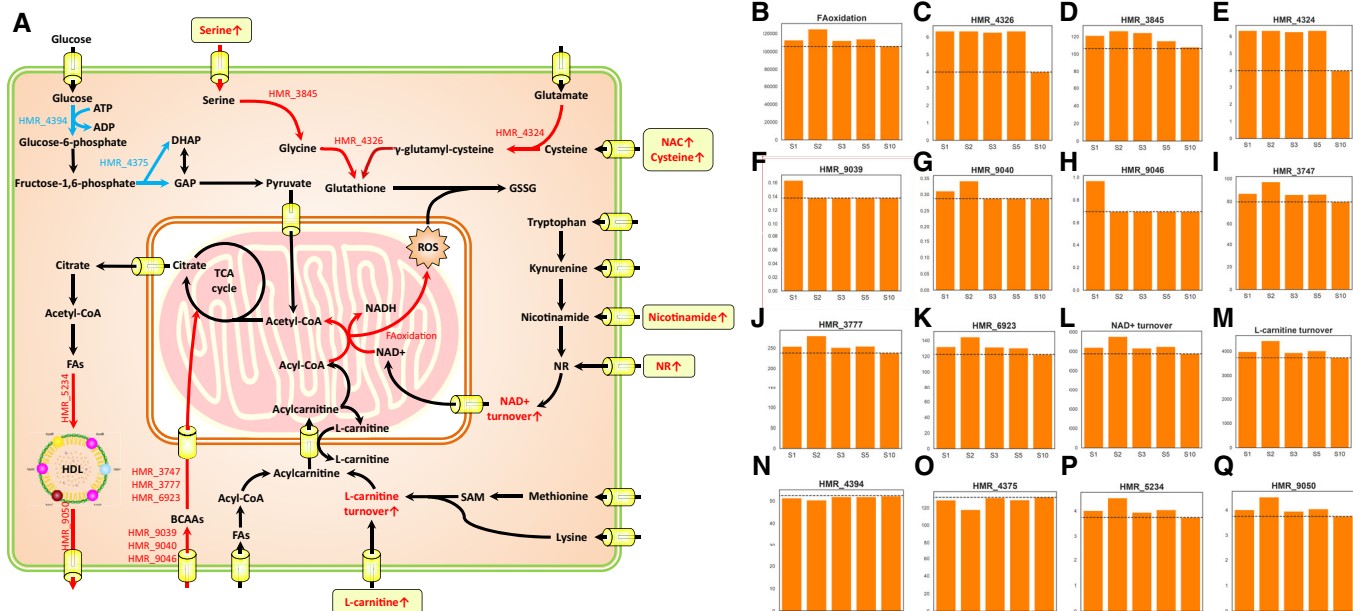

**Figure 5. The effect of supplementation on liver metabolism based on metabolic modeling.**

A    A summary of altered metabolic fluxes in liver by integrating metabolomics data accounting individual variations. Solid arrows indicate one or more metabolic reactions. Plasma metabolites increased due to metabolic cofactor supplementation are highlighted in rounded rectangle frames. The red/blue color of arrows and metabolite names indicate increased/decreased flux/pool with metabolic cofactor supplementation compared to control study.

B–Q    Changes in fluxes of key metabolic reactions highlighted in (A) compared to baseline fluxes (dashed line) based on personalized modeling, where "SX" indicating flux obtained by the modeling of subject number "X" as shown in Table EV13.

Interpolations of the plasma concentrations of each metabolic cofactor were constructed for every subject. The mean of the interpolations was used as the target concentration curve and the model was subsequently fitted to this curve. Once the model was fitted to each metabolic cofactor, the model was used to predict the resulting plasma concentration when subject to a twice-daily supplementation regimen. The individual dosages of the metabolic cofactors were adjusted to achieve a desired 100% increase in average (long term) plasma concentration without superseding safe doses for human consumption.

Interpolations of serine for each subject are displayed in Fig 6A and model fit to the mean plasma serine level is displayed in Fig 6B.

The model predicts that a twice-daily dose of 12.75 g serine may produce a desired long-term increase in mean plasma serine concentration of 100% (Fig 6C). Doses up to 400 mg/kg/day (around 25–30 g/day) have been studied in humans and shown to be safe (Garofalo *et al*, 2011).

Interpolations of L-carnitine for each subject are displayed in Fig 6D, and model fit to the mean plasma L-carnitine level is displayed in Fig 6E. The model predicts that a twice-daily dose of 8.3 g L-carnitine will produce a desired long-term increase in mean plasma L-carnitine concentration of 100% (Appendix Fig S6). However, since long-term supplementation studies for the safety of L-carnitine above 7 g/day have not been examined (Hathcock &

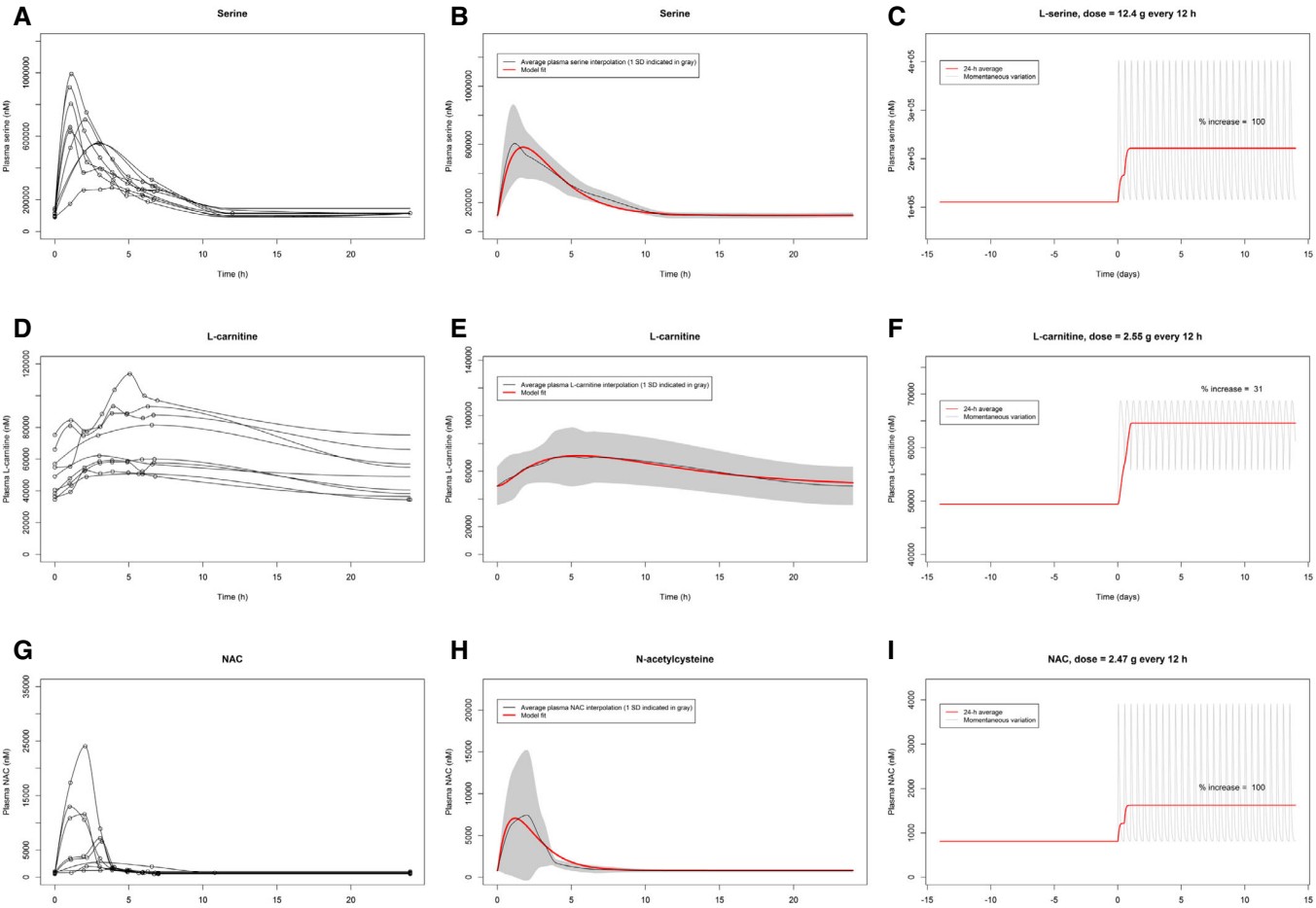

**Figure 6. Adjustment of the dose based on pharmacokinetic modeling.**

A  Plasma serine interpolations for each of the nine subjects.
B  Model fit to the target concentration curve (mean interpolation of plasma serine over 24 h) with standard deviation indicated in gray.
C  Predicted plasma serine during 2 weeks of simulated twice-daily supplementation: 24-h moving average value indicated in red and continuous/momentaneous indicated in gray.
D  Plasma L-carnitine interpolations for each of the nine subjects.
E  Model fit to the target concentration curve (mean interpolation of plasma L-carnitine over 24 h) with standard deviation indicated in gray.
F  Predicted plasma L-carnitine during 2 weeks of simulated twice-daily supplementation: 24-h moving average value indicated in red and continuous/momentaneous indicated in gray.
G  Plasma NAC interpolations for each of the nine subjects.
H  Model fit to the target concentration curve (mean interpolation of plasma serine over 24 h) with standard deviation indicated in gray.
I  Predicted plasma NAC during 2 weeks of simulated twice-daily supplementation: 24-h moving average value indicated in red and continuous/momentaneous indicated in gray.

 

Shao, 2006), the dose was reduced to 2.55 g twice daily (Fig 6F). This resulted in a long-term increase in mean plasma concentration of 31% (Fig 6F), which was considered a reasonable trade-off between risk of toxicity and increase in L-carnitine plasma concentration.

Interpolations of NAC for each subject are displayed in Fig 6G, and model fit to the mean plasma NAC level is displayed in Fig 6H. The model predicts that a twice-daily dose of 2.47 g NAC will produce a desired long-term increase in mean plasma NAC concentration of 100% (Fig 6I). A daily dosage of 4–6 g of NAC has been shown to be safe in humans (Hurd *et al*, 1996).

Regarding the dose adjustment of the NR, we used existing literature studies rather than performing additional modeling. Trammell *et al* (Trammell *et al*, 2016) supplemented 1 g/day NR for 7 days to one subject. Ten initial PBMC NAD$^+$ measurements were performed during the first 24 h, and one additional plasma NAD$^+$ measurement was performed after 7 days. Moreover, human data for NR pharmacokinetics in eight subjects with a non-compartmental analysis are available (Airhart *et al*, 2017). The study was a non-randomized, open-label PK study of eight healthy volunteers, 250 mg NR was orally administered on Days 1 and 2, and then up titrated to peak dose of 1,000 mg, twice daily on Days 7 and 8 (Airhart *et al*, 2017). On the morning of day 9, subjects completed a 24-h PK study after receiving 1,000 mg NR at $t = 0$. Whole-blood levels of NR, clinical blood chemistry, and NAD$^+$ levels were analyzed. Oral NR was well tolerated with no adverse events. Significant increases comparing baseline to mean concentrations at steady state were observed for both NR ($P = 0.03$) and NAD$^+$ ($P = 0.001$); the latter increased by 100%. Absolute changes from baseline to Day 9 in NR and NAD$^+$ levels correlated highly ($R^2 = 0.72$, $P = 0.008$). Considering the published experimental data, we decided to supplement 1 g of NR twice daily.

Based on the results of the ODE modeling and literature data, we concluded that supplementation of 12.35 g serine, 2.55 g of L-carnitine, 2.55 g of NAC, and 1 g of NR twice daily can be used for effective treatment of subjects. Considering that L-carnitine tartrate (salt form of L-carnitine) contains 67.6% of L-carnitine, 3.75 g L-carnitine tartrate twice daily can be used for effective treatment of subjects.

The modeling technique used to generate the graphs in Fig 6A–I, applied the pooled-data approach, where parameters were adjusted in the model to fit all subject data, simultaneously. The serine data were also analyzed using a population pharmacokinetic modeling approach with non-linear mixed-effect techniques. That analysis applied quantitative systems pharmacology (QSP) techniques, in which the physiologically meaningful parameters derived from our clinical data could be checked with available literature data. Human data for important parameters (such as endogenous serine production) were limited. Tsai *et al* (2008) provided volume and clearance values, but not separately from oral bioavailability. Wilcox *et al* (1985) provided additional data in psychotic and non-psychotic subjects. These data did allow bounding of model parameters. By using the limited literature data, we were able to better understand and quantify the physiology of serine uptake implied by modeling of our clinical results. The conclusions for dosing derived from both the pooled PK and QSP population techniques were identical. A more complete description of the QSP population modeling approach for serine will be reported elsewhere (preprint: Bosley *et al*, 2019).

## Discussion

We performed a calibration study by supplementing metabolic cofactors including L-serine, NAC, NR, and L-carnitine based on our previous study, where we proposed that the supplementation of such cofactors may boost uptake and oxidation of fatty acids in mitochondria and eventually decreased the amount of fat in liver (Mardinoglu *et al*, 2017). The rational of the supplementation is that liver itself has capacity to oxidize fat stored in liver in NAFLD patients. This has been justified in our recent study, where we showed that the liver fat content is decreased more than 40% by providing high fat and protein diet during 2 weeks in NAFLD patient (Mardinoglu *et al*, 2018c).

In our study, we performed a 7-day rat study to assess the tolerability of the combined metabolic cofactors including L-serine, NAC, NR, and L-carnitine and observed that none of the administered doses caused any detectable hematological, plasma chemistry or tissue effects. To study the potential immediate toxic effect of the combined metabolic cofactors in humans, we measured various clinically relevant biochemical parameters and observed no significant effect in a 5-day study.

We also generated untargeted metabolomics data and revealed the metabolic reprogramming in response to supplementation using a liver GEM. We found that the oxidation of fatty acids, *de novo* synthesis of GSH and catabolism of BCAAs was significantly increased whereas the consumption of glucose was significantly decreased after the supplementation of metabolic cofactors. Considering that these pathways are significantly associated with the progression of NAFLD, supplementation of metabolic cofactors may decrease the amount of fat in the liver of NAFLD patients.

Moreover, we obtained a very good agreement between the metabolic response in liver and the predicted effect of the metabolic cofactor supplementation. This highlights the great advantage of using natural metabolites as drugs, as the effects of natural metabolites may be more predictable. Compared to other types of drugs such as small molecules and proteins, we may have much more knowledge about the interactions of metabolites in human body based on GEMs and could predict the potential effect of such cofactors using tissue/cell-specific GEMs in a systematic way. Moreover, since the metabolites are already found in our bodies, we minimize the risk of some common problems in drug discoveries such as off-targeting and solubility, and we would expect much less side effect compared to small molecules. In this context, using a natural metabolic cofactor supplementation as medication could represent a promising direction in rational drug development especially in metabolism-related diseases including NAFLD, T2D, and cardiovascular diseases. Therefore, our study can be considered as a pivotal study indicating the power of this new direction in medicine.

We also applied pharmacokinetic modeling based on the dynamic metabolomics data. This allowed us to predict the long-term concentration response when supplementing the metabolites under a twice-daily supplementation regimen and it allowed determination of a physiologically feasible dosage. No predictions were made regarding variability of response between subjects. The predictions are also limited with regard to the phenotype of the test group (healthy subjects) and it is therefore possible that different groups would respond differently. Future supplementation studies

particularly related to traditional medicines could however use these predictions in order to design a supplementation regimen.

During pharmacokinetic modeling, a 100% increase was chosen as a partly arbitrary desired increase. If this increase however resulted in a dose larger than what was considered safe, the upper safe dose was used instead. An equally clinically effective lower dose than a 100% increase from baseline is not previously known; thus, we could not use such information when deciding the target dose. When supplementation starts the plasma concentration will, in the short-run, rise until a new long-term (approximate) steady state is reached. Depending on the compound, this new state will be reached in the order of days, after which the new long-term concentration will be maintained by further supplementation (continuing weeks to months or more).

The standard error of the estimated parameters $k_1$ and $k_2$ for the L-carnitine and NAC model is high (> 100% and > 1,000% respectively), meaning these parameters can assume different values and still fit the experimental data. This uncertainty may have been reduced if prior physiologic information on $k_1$ and $k_2$ had been available or if an alternative model would have been constructed. However, looking across the compounds serine, L-carnitine, and NAC, the $k_1$ parameter estimation is relatively similar (around three-fold difference in estimated parameter value) compared to the estimation of $k_2$ across compounds (around 20-fold difference in estimated parameter value). This points to a similar time dynamic across compounds in the stomach and small intestine and thus the difference in blood kinetics mainly being a result of the removal from the bloodstream rather than the influx into the bloodstream. This is physiologically plausible; thus, the final parameter estimations are arguably sensible.

In addition to fitting pharmacokinetic models to pooled data, we also applied population PK modeling to the serine data. The results from the naïve pooled-data and population approaches with respect to dose adjustment of serine were identical. This population PK model also exploited physiological data and known physiology to yield a QSP model, which matched not only the data from this calibration study, but also published data. The combination of population PK and QSP techniques provided more insight into (for example) endogenous generation of serine. These results will be presented elsewhere (preprint: Bosley *et al*, 2019).

Based on integrative analysis, we found that the oxidation of fatty acids, *de novo* synthesis of GSH, and catabolism of BCAAs were significantly increased whereas the consumption of glucose was significantly decreased after the supplementation of metabolic cofactors. Taken together, we observed that supplementation of these metabolic cofactors may provide a therapeutic strategy against NAFLD progression by promoting the uptake and oxidation of fat in mitochondria. Hence, we propose testing of metabolic cofactors in a randomized double-blind placebo-controlled human study to evaluate its long-term efficacy in NAFLD/NASH patients.

# Material and Methods

### Seven-day oral (gavage) tolerability study in the rat

We performed a rat study to provide a preliminary assessment of the tolerability of the combined metabolic cofactors including L-serine, NAC, NR, and L-carnitine tartrate (salt form of L-carnitine) at intended clinical doses and at levels 10- and 30-fold dose levels. All critical operations and methods were performed according to current standard operating procedures (SOPs) unless otherwise noted. The experiments were approved by the SWETOX, Swedish Toxicology Sciences Research Center Ethical Committee on Animal Experiments. The Study Director reviewed the analytical data for significant changes in any of the parameters.

The study comprised a 7-day repeated-dose phase with once-daily doses of the combined metabolic cofactors at three different dose levels to investigate the clinical tolerability by recordings of clinical signs and analysis of clinical pathology parameters. Blood samples were taken for plasma exposure analysis at expected $T_{max}$. Livers and kidneys were isolated and fixated at necropsy for histopathological analyses and analysis of organ exposure.

The oral route is the intended clinical administration route. Details of numbers of animals and dose levels are included in Appendix Table S1. Nine female rats were dosed once daily for 7 days. Animals were administered combined metabolic cofactors formulated in tap water per orally (gavage) once daily for seven consecutive days (Day 1–7). Three groups of three female rats/ group (Groups 1, 2, and 3) were dosed with the test formulation at the dose levels described in Appendix Table S1. In Group 3, the dose administered was not well tolerated at Days 1 and 2, and on Day 3, the dose volume was reduced from 10 to 6.7 ml/kg/day, resulting in a reduction of the high dose with on third for Days 3–7. The animals were dosed in the morning (before noon). Animals were allowed to recover for at least 20 h before any subsequent doses will be administered (Appendix Table S2).

All animals were killed for scheduled necropsy on Day 8 by exsanguinations under isoflurane and oxygen anesthesia. For scheduled necropsy, the thoracic and abdominal cavities and contents were examined. Macroscopic abnormalities were recorded. Tissues listed below were collected and fixed for further potential analysis of tissue changes and tissue exposures. Samples of the following tissues (kidneys, liver, intestine—jejunum, intestine—ileum, intestine—colon, intestine—cecum, intestine—rectum, stomach/duodenum) were taken at necropsy for possible future microscopic pathology. Tissues were fixed and preserved in 4% formaldehyde, for possible future processing and histological analysis.

**Blood and urine sample collection for hematology and plasma chemistry analysis**

| Animals bled | All animals |
|---|---|
| Blood collection site and schedule | Blood for scheduled hematology and plasma chemistry was taken during termination at necropsy from the orbital plexus, carotid artery or according to Swetox GLP-SOP-07. |
| Blood sample volume and anticoagulant | Plasma chemistry: at least 0.25 ml at necropsy ml (lithium heparin). Hematology: at least 20 µl (EDTA). |
| Analyses of samples | Blood was analyzed using Exigo (Hematology) and Vetscan (Plasma Chemistry) equipment by the Responsible technician for Pathology according to user manuals or local SOPs. |

(continued)

| Animals bled | All animals |
|---|---|
| Information to be provided with the samples | The samples were transferred for analyses with the following sample information. Sample list stating actual sampling times. All samples were labeled with study number, sample number, and animal number and day/time point. |
| Hematology sample procedures and analyses | Blood was analyzed within 60 min of sampling. On the morning of analysis, the Exigo analyzer was checked against a reference sample provided by the manufacturer, and a control sample analysis cycle was performed.<br>For each study sample, SIG-41, animal ID, sample number were manually entered on the analyzer.<br>The analyzer was set to automatic printout, and each individual sample printout was collected and saved in the study folder. The data are presented in Appendix Table S3. |
| Plasma chemistry sample procedures | Plasma chemistry was analyzed as whole blood. The Vetscan analyzer was calibrated with calibration samples with known plasma levels of each parameter before start of analysis of study samples.<br>For each study sample, SIG-41, animal ID, sample number were manually entered on the analyzer.<br>The analyzer was set to automatic printout and each individual sample printout was collected and saved in the study folder. The data are presented in Appendix Table S4. |

## Human calibration study

We performed a 5-day calibration study by recruiting nine healthy male subjects without any medication (age 26–36 years, BMI 19.4–34.5 kg/m$^2$). After 21 months, we asked the subjects involved in the study to run a follow-up control study and five of the subjects responded positively. We recruited another five healthy subjects and run a 1-day control study with 10 male subjects without any medication (age 26–36 years, BMI 19.8–35.8 kg/m$^2$) following the same study design in the metabolic cofactor supplementation study. The inclusion criteria are that the patients should be healthy without any medication, no smokers, and no obesity. The exclusion criteria are that if the patient has known disease or obesity. Informed consent was obtained from all subjects involved in this study, and the experiments conformed to the principles set out in the WMA Declaration of Helsinki and the Department of Health and Human Services Belmont Report. The study was approved by the Ethics Committee at the University of Gothenburg, and it has been submitted to https://ClinicalTrials.gov with the identifier: NCT03838822.

## Detection of plasma metabolite levels

Measurement of plasma levels of metabolites was performed using LC-MS. Briefly, the liquid chromatography–tandem mass spectrometry (LC-MS/MS) platform was based on a Waters ACQUITY ultraperformance liquid chromatography (UPLC) system and a Thermo-Finnigan LTQ mass spectrometer operated at nominal mass resolution, which was equipped with an electrospray ionization (ESI) source and a linear ion trap (LIT) mass analyzer. The sample extract was dried and then reconstituted in acidic or basic LC-compatible solvents, each of which contained 12 or more injection standards at fixed concentrations. One aliquot was analyzed using acidic positive ion-optimized conditions and the other was analyzed using basic negative ion-optimized conditions in two independent injections using separate dedicated columns (Waters UPLC BEH C18–2.1 × 100 mm, 1.7 μm). Extracts reconstituted in acidic conditions were gradient eluted using water and methanol containing 0.1% formic acid, whereas the basic extracts, which were also eluted using water/methanol, contained 6.5 mM ammonium bicarbonate. The MS analysis alternated between MS and data-dependent MS/MS scans using dynamic exclusion, and the scan range was from 80 to 1,000 $m/z$. To identify the significant differences of plasma metabolites compared to baseline (time T0), one-way ANOVA was employed for each time points. Furthermore, to study associations between plasma levels of metabolites supplemented by the cocktail and untargeted metabolites, Spearman correlation was used.

## Detection of plasma inflammation protein marker levels

Plasma inflammation proteins were analyzed using a multiplex proximity extension assay (Olink Bioscience, Uppsala, Sweden). The kit provides a microtiter plate for measuring 92 protein biomarkers in each plasma sample. Each well contains 96 pairs of DNA-labeled antibody probes. Samples were incubated in the presence of proximity antibody pairs tagged with DNA reporter molecules. When the antibodies pair bounds to their corresponding antigens, the corresponding DNA tails form an amplicon by proximity extension, which can be quantified by high-throughput real-time PCR. Briefly, 1 μl of each sample was mixed with 3 μl probe solution containing a set of 92 protein target-specific antibodies conjugated with distinctive DNA oligonucleotides. The mixture was incubated overnight at 4°C and then added with 96 μl extension solution containing extension enzyme and PCR reagents. The generated fluorescent signal allows the quantification of the protein using BioMark™ HD System, Fluidigm Corporation. To minimize inter- and intra-run variation, the data are normalized using both an internal control (extension control) and an interplate control and then transformed using a pre-determined correction factor. The pre-processed data were provided in the arbitrary unit Normalized Protein eXpression (NPX) on a log$_2$-scale, which were then linearized by using the formula 2NPX. A high NPX presents high protein concentration. Limit of detection (LOD) for each protein was defined as three standard deviations above the background. In this study, Olink Inflammation panel has been used.

## Statistics used for metabolomics and proteomics data analysis

Both plasma metabolomics and protein inflammation markers dataset were pre-processed and analyzed in the same manner. First, all data points in both supplementation and control studies were normalized into fold changes by comparing to their baseline value which is time 0. After the pre-processing, data were then respectively compared at each time point with their baseline data using one-way ANOVA in order to determine which metabolites/proteins were showing statistically significant difference changes over time

(FDR < 5%). In addition, data were also respectively compared at each time point between supplementation and control studies using one-way ANOVA to determine which metabolites/proteins were showing statistical alteration in response to the metabolic cofactor supplementation. Correlation analysis (Spearman) was also performed between each metabolite with the supplemented metabolites (serine, L-carnitine, and *N*-acetylcysteine for both conditions and nicotinamide in the cocktail). In all analyses, missing values were removed in a pairwise manner. ANOVA and correlation analyses were performed using SciPy package in Python.

### Genome-scale metabolic modeling (GEM) of liver tissue metabolism

First, we generate a GEM for the reference state of liver in this study. The liver GEM from our previous study is used as a starting point and 62 additional exchange reactions were added. An ATP hydrolysis reaction is also added to account for the resting energy expenditure, and a low bound of 1,212.4 is added to it which is calculated from previous experimental measurements (15 kcal/liver/h). The constraints for internal reactions are set to +/− infinity if they are reversible, or 0 to infinity if they are irreversible. The upper and lower bounds for the exchange reactions of cocktail substances are respectively set to 0.1 and 1 to force the uptake. The exchange of nicotinamide and cysteine is also set to be uptake only since they are closely related to the cocktail supplementation. Constraints for exchange reactions are set based on previous studies wherever possible. For the ones that has not reported, the constraints are set based on metabolomics data. Specifically, uptake and secretion for all metabolites are set to 1 except for irreversible reactions whose lower bounds have to be 0, and some special constraints are set to several reactions such as $O_2$ uptake, $H_2O$ exchange, etc., where the uptake should be unlimited. The objective function for simulation is set as the total fatty acid oxidation flux in mitochondria to reduce the solution space. This only counts the fatty acid oxidation reactions in mitochondria, and each of them is weighted by the total number of FADH2 and NADH produced. In addition, the turnover flux of $NAD^+$ and L-carnitine was calculated by summing up the total absolute flux of them in all compartments.

An adapted RMetD was used to simulate the personalized metabolic flux distribution with cocktail supplementation. First, we sampled 1,000 times for the reference state model and used the average flux as representative for the reference metabolic activity in liver. Then, the fold changes for each metabolite are used to change the upper/lower bounds of the corresponding exchange reaction. For instance, if a metabolite is uptake by the model and its plasma level is decreased with a fold change of two with cocktail supplementation, it indicates that uptake of this metabolite is increased. In this case, the lower bound of it will be set as its reference flux, and its upper bound is set as two times of its reference flux. On the contrary, if a metabolite is secreted by the model and its plasma level is decreased with a fold change of 2, the upper bound of it will be set as its reference flux, and its lower bound is set as its reference flux divide by 2. After the change of constraints according to the individualized changes, we again calculated the flux distribution of each personalized GEM with 1,000 times random sampling and obtained the personalized flux distribution with cocktail supplementation by averaging the sampled fluxes.

### Prediction of clinical doses using an ordinary differential equation model

The following model was used for simulation of the substances serine, L-carnitine, and *N*-acetyl-cysteine:

$$\frac{dA}{dt} = -k_1 \cdot A$$

$$\frac{dB}{dt} = k_1 \cdot A - k_1 \cdot B$$

$$\frac{dC}{dt} = k_1 \cdot B - k_2 \cdot (C - C_0)$$

where $A$ represents the stomach, $B$ represents the small intestine, and $C$ represents the blood. The constant $C_0$ represents the baseline concentration of the compound of interest. The parameter $k_1$ was used both as the rate constant for the stomach and the small intestine since it reduced the degrees of freedom for the model but provided a time delay which better fitted the experimental data. Alternatively, one compartment representing both the stomach and the small intestine could have been used; however, this produced worse fit to the experimental data.

### Parameter values and standard errors for serine model

|  | Parameter value | RSE (%) |
|---|---|---|
| $k_1$ | 0.60916 | 21.3 |
| $k_2$ | 18.16688 | 21.3 |

### Parameter values and standard errors for L-carnitine model

|  | Parameter value | RSE (%) |
|---|---|---|
| $k_1$ | 0.20350 | 168.8 |
| $k_2$ | 1.67995 | 164.2 |

### Parameter values and standard errors for NAC model

|  | Parameter value | RSE (%) |
|---|---|---|
| $k_1$ | 0.29173 | 2,150.3 |
| $k_2$ | 28.32394 | 1,516.6 |

## Data availability

The datasets and computer code produced in this study are provided as Computer Code EV1.

**Expanded View** for this article is available online.

## Acknowledgements

This work was financially supported by the Knut and Alice Wallenberg Foundation, Swedish Research Foundation, and Swedish Heart-Lung Foundation. The authors would like to thank ChromaDex (Irvine, CA, USA) for providing NR for this study.

## Author contributions

CZ performed the computational analysis and analyzed the clinical data together with EB, AT, HT, AL, MAr, GB, RB, MO, KJ, JTK, WK, JN, M-RT, MU, MAd, JBos, AM. JBor and H-UM coordinated the generation of the clinical data. JBor, MS, and P-OB generated the metabolomics data. CZ and AM wrote the paper and all authors were involved in editing the paper.

## Conflict of interest

AM, JB, and MU are the founder and shareholders of ScandiBio Therapeutics and ScandiEdge Therapeutics. The other authors declare that they have no conflict of interest.

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
