## [Review Process File · Molecular Systems Biology]

The acute effect of metabolic cofactors supplementation, a potential therapeutic strategy against non-alcoholic fatty liver disease

Cheng Zhang, Elias Bjornson, Muhammad Arif, Abdellah Tebani, Alen Lovric, Rui Benfeitas, Mehmet Ozcan, Kajetan Juszcak, Woonghee Kim, Jung Tae Kim, Gholamreza Bidkhori, Marcus Ståhlman, Per-Olof Bergh, Martin Adiels, Hasan Turkez, Marja-Riitta Taskinen, Jim Bosley, Hanns-Ulrich Marschall, Jens Nielsen, Mathias Uhlen, Jan Boren and Adil Mardinoglu.

Review timeline:	Manuscript submitted:	7 th February 2020
	Editorial Decision:	2 nd March 2020
	Revision received:	4 th March 2020
	Accepted:	10 th March 2020

Editor: Maria Polychronidou

Transaction Report:

The reviewers' comments and authors' responses are not available with this article, as the initial review process took place with another journal.

1st Editorial Decision

2nd March 2020

Thank you again for submitting your work to Molecular Systems Biology. We have now heard back from the two referees who agreed to evaluate your study. Reviewer #1 and reviewer #2 are the previous reviewers #2 and #3. Both reviewers mention that their concerns have been satisfactorily addressed and are supportive of publication.

Before we formally accept the study for publication, we would ask you to address some editorial issues listed below.

REFEREE REPORTS

Reviewer #1:

The authors have well addressed my prior critiques. I have no further concerns with the paper.

Reviewer #2:

The authors have addressed my comments.

1st Revision - authors' response

4th March 2020

The Authors have made the requested editorial changes.

Accepted

10th March 2020

Thank you for performing the requested changes. We are now satisfied with the modifications made and I am pleased to inform you that your paper has been accepted for publication.

Corresponding Author Name: Adil Mardinoglu

Manuscript Number: MSB-20-9495R